# Toward Better Pedestrian Trajectory Predictions: The Role of Density and Time-to-Collision in Hybrid Deep-Learning Algorithms

**DOI:** 10.3390/s24072356

**Published:** 2024-04-08

**Authors:** Raphael Korbmacher, Antoine Tordeux

**Affiliations:** Department for Traffic Safety and Reliability, University of Wuppertal, 42119 Wuppertal, Germany; tordeux@uni-wuppertal.de

**Keywords:** pedestrian trajectory prediction, deep learning, pedestrian trajectory dataset, density-based classification, collision avoidance

## Abstract

Predicting human trajectories poses a significant challenge due to the complex interplay of pedestrian behavior, which is influenced by environmental layout and interpersonal dynamics. This complexity is further compounded by variations in scene density. To address this, we introduce a novel dataset from the Festival of Lights in Lyon 2022, characterized by a wide range of densities (0.2–2.2 ped/$m^{2}$). Our analysis demonstrates that density-based classification of data can significantly enhance the accuracy of predictive algorithms. We propose an innovative two-stage processing approach, surpassing current state-of-the-art methods in performance. Additionally, we utilize a collision-based error metric to better account for collisions in trajectory predictions. Our findings indicate that the effectiveness of this error metric is density-dependent, offering prediction insights. This study not only advances our understanding of human trajectory prediction in dense environments, but also presents a methodological framework for integrating density considerations into predictive modeling, thereby improving algorithmic performance and collision avoidance.

## 1. Introduction

The challenge of predicting pedestrian trajectories has emerged as a pivotal challenge in recent years. This surge in interest is largely attributed to the profound implications it holds for autonomous vehicle navigation [1], service robot deployment [2], and the strategic planning of infrastructure and mass gatherings [3]. In addressing these intricate challenges, researchers have traditionally employed physics-based (PB) models to simulate and understand pedestrian behavior. These models have been instrumental in dissecting collective phenomena and enhancing our understanding of pedestrian dynamics, particularly in high-density contexts relevant to crowd management and evacuation strategies [3]. However, the landscape of pedestrian trajectory prediction has witnessed a paradigm shift over the last decade with the advent and integration of deep learning (DL) algorithms [4]. Despite the opaqueness of these models in terms of interpretability, their superiority in mirroring observed trajectories has been markedly pronounced, especially when juxtaposed with their PB counterparts [5]. Nonetheless, it is important to acknowledge that the domains of applicability for PB and DL models do not entirely overlap. While PB models excel in the realm of high-density simulations, providing insights into collective behavior, DL algorithms predominantly thrive in low-density environments where individual pedestrian movements are characterized by a greater degree of freedom and intricate long-range interactions [4]. This paper introduces a novel, real-world pedestrian trajectory dataset, gathered during the Festival of Lights in Lyon. Field pedestrian trajectory datasets are typically gathered from low-density situations. In contrast, this dataset captures the nuanced dynamics of pedestrian movements across a large spectrum of density levels. It ranges from sparse crowds observed during show moments to the densely packed throngs seen after the event. Utilizing this dataset, we train DL algorithms, including Long Short-Term Memory (LSTM) networks and Generative Adversarial Networks (GAN). Our methodology is underscored by a novel approach: we harness situational classification predicated on crowd density to refine our models’ learning process. This two-stage process, which initially classifies the scene based on density and then predicts the trajectories, not only bolsters the efficiency of our models but also substantially elevates the precision of trajectory predictions. This improvement is demonstrated by comparative analyses with traditional DL algorithms and PB models.

Another challenge of trajectory prediction that we face in this paper is overlapping and colliding of predicted trajectories [6]. To tackle that problem, we integrate a time-to-collision (TTC) [7] term into the loss function of the algorithms. A parameter, λ, is utilized to modulate the TTC’s influence on the training. Our empirical research uncovers a significant relationship between the optimal λ values and the density levels, highlighting the intricacies of pedestrian behavior across different densities.

The remainder of this paper is organized as follows: first, we review related studies in Section 2. Then, our novel dataset is presented in Section 3 and the methodology for the empirical work is proposed in Section 4. The results are shown in Section 5. Finally, Section 6 includes a discussion of the results and an outlook on future works.

## 2. Related Work

The domain of pedestrian trajectory prediction is multifaceted, drawing insights from various disciplines and methodologies. The two main stream are the physics-based models and data-based algorithms [4]. PB models have been the cornerstone of understanding pedestrian dynamics, especially in high-density scenarios. The Social Force model (SF), introduced by Helbing and Molnar [8], exemplifies this approach, simulating pedestrian movement by balancing attractive and repulsive forces. Other famous PB models are the Optimal Reciprocal Collision Avoidance (ORCA) from Van den Berg et al. [9] or the cellular automata model from Burstedde et al. [10]. However, these models are not without their challenges, particularly when it comes to encapsulating the full range of crowd behavior [4]. For more PB models, see reviews such as [11,12,13].

In pursuit of addressing these limitations, the research frontier has gradually shifted towards data-driven methodologies. Notably, the past decade has witnessed a burgeoning interest in DL approaches. Pioneering works such as the Social LSTM from Alahi et al. [5] introduce the use of Recurrent Neural Networks (RNN), specifically LSTM networks, in conjunction with a novel concept known as Social Pooling. This innovative approach incorporates neighbouring information, thereby enriching the model’s contextual understanding. This social concept was further enhanced by Gupta et al. [14] through Social GAN, where the generative adversarial framework allowed for the generation of multiple plausible future paths, addressing the inherent uncertainty in human movement. Alternative methods employed for social predictions include attention mechanisms [15], graph-based approaches [16], and the utilization of relative coordinates [17]. Additionally, deep learning architectures such as Convolutional Neural Networks [18,19] and Transformers [20] have been applied in trajectory prediction tasks.

A pivotal aspect of this paper is the innovative classification of trajectory scenes based on crowd density prior to prediction. To the best of our knowledge, this approach is a novel paradigm, potentially owing to the scarcity of high-density, real-world pedestrian trajectory datasets. Xue et al. [21] predicted pedestrian destinations using bidirectional LSTM classification. This involves an additional classification stage to distinguish between possible destinations of pedestrians. They classify the route manually into four distinct categories. In another paper from the authors [22], the classification was based on a clustering algorithm. Kothari et al. [23] categorized pedestrian trajectories based on the nature of interactions observed, identifying behaviors such as collision avoidance, leader–follower dynamics, and grouping behavior. An alternative methodology involves classifying trajectories based on individual pedestrian characteristics. Papathanasopoulou et al. [24] concentrated on attributes such as age, gender, height, and speed to inform their classification. A second cornerstone of this work is the seamless integration of a PB concept, TTC, into the loss function of DL algorithms. This synthesis of PB principles and DL models is not an isolated endeavor. Alahi et al. [5] and Khadka et al. [25] utilized simulated data from PB models for training DL algorithms. Antonucci et al. [26] embedded a PB model directly into the DL architecture. Furthermore, the works of Silvestri et al. [27] and Kothari et al. [28] stand out for their use of PB principles within the loss function to eliminate unrealistic predictions.

## 3. The Dataset

With a growing interest in data-based methods, the significance of pedestrian trajectory data has been elevated in recent research. This area has seen a proliferation of datasets published by researchers, which can be categorized into field data and experimental data obtained in laboratory conditions. In the field studies, real-world settings are employed where individuals, unaware of their participation in a study, navigate through various scenarios. Famous field datasets are the ETH [29] and UCY [30] datasets, which are widely used in the machine learning community. Originating from surveillance videos, these datasets capture pedestrians scene of low density (0.1–0.5 ped/$m^{2}$). The GLOW dataset from Eindhoven [31], a dataset used for route choice analysis, contains trajectory scenes of higher densities, but only for short length trajectories. Other field datasets are the Stanford Drone Dataset [32], the Grand Central Station Dataset [33], and the Edinburgh Informatics Forum Dataset [34]. None of these have densities above 0.2 ped/$m^{2}$. In the following, we will present a field dataset with pedestrian densities between 0.2 and 2.2 ped/$m^{2}$.

The data were collected at Lyon’s Festival of Lights. The event runs for four days from 7 p.m. to 11 p.m., and attracts millions (2 million in 2022) of visitors each year. Key attractions are light shows at Place des Terreaux and Place Saint-Jean. We have installed cameras at the Place des Terreaux to film the area, which is represented by the red rectangle in Figure 1.

In Figure 1a, the entirety of Place des Terreaux is depicted. The red box on the right-hand side delineates our designated tracking region. Figure 1b offers an aerial perspective of this same area. This designated zone measures 9 m in length and 6.5 m in width. On average, we concurrently tracked 55 pedestrians, resulting in a mean density of 0.95 ped/$m^{2}$. The distribution of pedestrian density exhibited significant variability. During the light show, the majority of pedestrians congregated in the central area of the square, remaining largely stationary. Consequently, the pedestrian density within our tracking zone was relatively low. However, when the show concluded—which consistently lasts approximately 9 min—the crowd dynamics shifted dramatically, as most individuals sought to exit towards another event. In this transition phase, the density within our tracking corridor surged, often exceeding 120 pedestrians moving simultaneously. For video calibration, we meticulously established nine calibration points, ensuring the precise tracking of pedestrian trajectories using the PeTrack software [35]. Throughout our study, we recorded 5195 individual trajectories, which averaged a duration of 12.38 s and a mean velocity of 0.62 m/s. The pedestrian flow changes between mostly unidirectional flow, after the light show and bidirectional flow during the show. More informations about the data can be found in the Appendix A. For the training and testing of the algorithms, we need trajectories of a minimal length of 7 s (see Section 4.1). Because many trajectories are more than 14 s long, they can be used more than once. In total, we obtained 7450 trajectories for training and testing.

## 4. Methodology

### 4.1. Overview

For predicting pedestrian trajectories, we have $i^{th}$ pedestrian in a scene represented by image coordinates $\left(x_{t}^{i},y_{t}^{i}\right)$ for each time instant t=k·dt, with k∈N and a time step of dt=1/3 s. The observed positions from $t=T_{1}$ to $t=T_{obs}$ is taken as input and the aim is to predict future trajectories from $t=T_{obs}+dt$ to $t=T_{pred}$. Every scene involves a primary pedestrian and their neigbours over the timespan $T_{1}$ to $T_{pred}$. A neigbours is a pedestrian whose position at $T_{1}$ is closer to the position of the primary pedestrian than a radius r=5 m. Our dataset has a framerate of three observation for each second. We choose input trajectories of 9 observations (3 s) and want to predict 12 timesteps (4 s).

The predicted trajectory of all primary pedestrians are evaluated on two commonly utilized Euclidean distance metric and a collision metric. In the first distance-based metrics, called average displacement error (ADE) [29], the distance between the predicted trajectory and the ground truth trajectory is measured at any time step *t*:(1)$$\mathrm{ADE}=\frac{1}{NT}\sum_{i=1}^{N}\sum_{t=1}^{T}∥{\hat{x}}_{i}\left(t\right)-x_{i}\left(t\right)∥.$$
where $x_{i}\left(t\right)$ is the actual position of the $i^{th}$ pedestrian at time *t* and ${\hat{x}}_{i}\left(t\right)$ is the predicted position. The Euclidean distance is denoted as ∥·∥. The second distance-based metric, called the final displacement error (FDE) [30], displays the distance between the final point $t=T_{pred}$ of the predicted trajectory and the ground truth trajectory:(2)$$\mathrm{FDE}=\frac{1}{N}\sum_{i=1}^{N}∥{\hat{x}}_{i}\left(T\right)-x_{i}\left(T\right)∥.$$

These distance-based metrics are widely used in pedestrian trajectory predictions for their effectiveness in quantifying the goodness-of-fit. However, repulsive forces, which are pivotal in shaping interactions between pedestrians, are not taken into account [36]. Consequently, these metrics do not account for potential overlaps or collisions between pedestrians. Therefore, the collision metric is used to enhance the evaluating process:(3)$$\mathrm{COL}=\frac{1}{\left|S\right|}\sum_{\hat{X}\in S}COL\left(\hat{X}\right),$$
with:(4)$$COL\left(\hat{X}\right)=\min\left(1,\sum_{t=1}^{T}\sum_{i=1}^{N}\sum_{j>i}^{N}\left[\left|\right|{\hat{x}}_{i}\left(t\right)-{\hat{x}}_{j}\left(t\right)\left|\right|\leq 2R\right]\right).$$
where *S* includes all scenes in the test set, $\hat{X}$ represents a scene prediction containing *N* agents, and ${\hat{x}}_{i}$ is the prediction of the position of the agent *i* over the prediction time of *T*, while [·] is the Iverson bracket.
(5)$$\left[P\right]=\left\{\begin{array}{cc} 1 & \mathrm{if}\;P\;\mathrm{is}\;\mathrm{true},\\ 0 & \mathrm{otherwise}. \end{array}\right.$$

This metric counts a prediction as a collision when a predicted pedestrian trajectory intersects with neighboring trajectories, thus indicating the proportion of predictions where collisions occur. A vital factor in this calculation is the chosen pedestrian size *R*. An increase in *R* will likewise increase the number of collisions. For calculating the collision metrics, we use a radius R=0.2 m.

### 4.2. Prediction Approaches

In the subsequent trajectory predictions, various trajectory prediction approaches, ranging from traditional PB models to modern DL algorithms, are chosen as benchmarks for comparison with our two-stage approach. We present the results of the Constant Velocity model (CV) and SF model [8] as well as the results of a Vanilla LSTM, the Social LSTM (SLSTM) [5], and the Social GAN [14]. These approaches, characterized by their diverse features, are commonly selected for comparison and serve as benchmarks that must be surpassed. The position and velocity of the $i^{th}$ pedestrian are denoted as $x_{i}\in R^{2}$ and $v_{i}\in R^{2}$, respectively. For a system of *N* pedestrians, the position and velocity vectors, $x=(x_{1},\ldots ,x_{N})$ and $v=(v_{1},\ldots ,v_{N})$, have dimensions of 2N. All variables, including x(t) and $x_{i}\left(t\right)$, are functions of time *t*.

#### 4.2.1. Constant Velocity Model

The CV model assumes pedestrian velocities remain unchanged over time. It serves as a baseline for more complex models. The future position of a pedestrian is predicted as:(6)$$x_{i}\left(t+t_{p}\right)=x_{i}\left(t\right)+t_{p}v_{i}\left(t\right),\;\forall t_{p}\in \left[0,T_{p}\right].$$

#### 4.2.2. Social Force Model

Introduced by Helbing and Molnar [8], the SF model treats pedestrians as particles influenced by forces. Within this framework, the model calculates acceleration based on the cumulative effect of three distinct forces, as delineated in Equation (Equation 7):(7)$$m_{i}\frac{dv_{i}}{dt}=m_{i}\frac{v_{i}^{0}-v_{i}}{\tau }+\sum_{j\neq i}\nabla U\left(x_{j}-x_{i}\right)+\sum_{W}\nabla V\left(x_{W}-x_{i}\right)$$
where $m_{i}$, $v_{i}$, and $v_{i}^{0}$ signify the mass, current velocity, and desired velocity of pedestrian *i*, respectively. The term $\nabla U(x_{j}-x_{i})$ represents the repulsive force from other pedestrians, while $\nabla V(x_{W}-x_{i})$ indicates the repulsive force from obstacles. The potential functions U(d) and V(d) are given by:(8)$$U\left(d\right)=ABe^{-|d|/B},\;A,B>0\;\mathrm{and}\;V\left(d\right)=A^{\prime }B^{\prime }e^{-\left|d\right|/B^{\prime }},\;A^{\prime },B^{\prime }>0$$
where $A,A^{\prime },B,B^{\prime }>0$ are interaction parameters of the social force model.

The first term of Equation (Equation 7) signifies the driving force experienced by the $i^{th}$ pedestrian. This force propels the individual towards their desired speed and direction within a relaxation time τ>0. The second term encapsulates the summation of social forces, originating from the repulsive effects as pedestrians endeavour to maintain a comfortable distance from one another. The third term accounts for the aggregate interaction forces between pedestrian *i* and various obstacles. While the CV model has no parameter, the SF model has three parameters, i.e., preferred velocity, interaction potential, and reaction time, which can be optimized to get accurate predictions.

#### 4.2.3. Vanilla LSTM

LSTM networks, a class of RNN designed to learn long-term dependencies, have proven effective in handling sequential data, particularly for time series prediction tasks. Introduced by Hochreiter and Schmidhuber [37], LSTMs address the vanishing and exploding gradient problems common in traditional RNNs, making them suitable for complex sequence modeling tasks such as trajectory prediction. The vanilla LSTM model considers historical trajectories to predict future positions.
(9)$$x_{i}\left(t+t_{p}\right)=x_{i}\left(t\right)+\mathrm{LSTM}\left(t_{p},\left(x_{i}\left(t-t_{o}\right),\;t_{o}\in \left[0,T_{o}\right]\right)\right),\;\forall t_{p}\in \left[0,T_{p}\right].$$
where $x_{i}\left(t+t_{p}\right)$ predicts the future trajectory of a pedestrian *i* at time $t+t_{p}$, based on its past positions $x_{i}\left(t-t_{o}\right)$, over an observation window $t_{o}\in \left[0,T_{o}\right]$.

#### 4.2.4. Social LSTM

LSTM networks have demonstrated effective performance in sequence learning tasks. One such task, the prediction of pedestrian trajectories, presents the challenge that the trajectory of a pedestrian can be significantly influenced by the trajectories of surrounding pedestrians. The number of these neighboring influences can fluctuate widely, especially in densely crowded environments [38].

Enhancing the LSTM framework, the SLSTM by Alahi et al. [5] incorporates a social pooling layer, enabling the model to consider the influence of neighboring pedestrians explicitly. This is a key distinction from the Vanilla LSTM, reflecting the model’s capacity to capture social interactions:(10)$$x_{i}\left(t+t_{p}\right)=x_{i}\left(t\right)+\mathrm{SLSTM}\left(i,t_{p},\left(x\left(t-t_{o}\right),\;t_{o}\in \left[0,T_{o}\right]\right)\right),\;\forall t_{p}\in \left[0,T_{p}\right].$$
where *x* is the vector of positions of the neighboring pedestrians. In this formulation, the inclusion of the index *i* and the collective pedestrian state *x* emphasizes the model’s attention to the surrounding pedestrians’ trajectories, making it adept at handling complex social behaviors in dense scenarios.

#### 4.2.5. Social GAN

Another approach we take into account is the Social GAN (SGAN) introduced by Gupta et al. [14]. This model extends traditional approaches by incorporating GANs to predict future trajectories. GANs, conceptualized by Goodfellow et al. [39], consist of two competing networks: a Generator, which generates data samples, and a Discriminator, which evaluates the authenticity of the samples against real data. SGAN leverages this architecture to generate plausible future trajectories of pedestrians, addressing the complex dynamics of pedestrian movement in crowded spaces. A key feature of the SGAN model is its pooling mechanism, which processes the relative positions of pedestrians to each other. This mechanism is crucial for understanding the social interactions and dependencies among individuals in crowded environments
(11)$$x_{i}\left(t+t_{p}\right)=x_{i}\left(t\right)+\mathrm{SGAN}\left(i,t_{p},\left(x\left(t-t_{o}\right),\;t_{o}\in \left[0,T_{o}\right]\right)\right),\;\forall t_{p}\in \left[0,T_{p}\right].$$

### 4.3. Two-Stage Process

The foundation of our innovative classification framework lies in its capacity to predict trajectories across varying density levels, marking a departure from traditional models that typically utilize a single algorithm to process a wide array of scenarios within a dataset. Our strategy entails segmenting the dataset according to the density of each scene, thereby generating distinct subsets. At the inception of our methodology, we establish well-defined criteria for classification. This process is underpinned by two distinct methodologies: a statistical analysis and a review of existing literature. The results of this clustering process are depicted in Figure 2. These figures visually represent each dataset item as a point measurement each second, with the left side of Figure 2 illustrating points based on their average density:(12)$$\rho \left(t\right)=\frac{N(t)}{A},$$
and average velocity:(13)$$\bar{v}\left(t\right)=\frac{1}{N(t)}\sum_{i\in S(t)}v_{i}\left(t\right),$$
for the K-Means clustering. On the right side, the clustering is carried out by using the Agglomerative Hierarchical Clustering algorithm (AHC).

We can see clear vertical colour switch’s of the points at densities around 0.7, 1.1, and 1.6 ped/$m^{2}$ for both cluster algorithms. Remarkably, without presetting the number of clusters or explicitly focusing on density levels, the K-Means and AHC algorithms autonomously reveal density-dependent clustering. The delineation of clusters and their boundary values align closely with those identified in the literature. Stefan Holl [40] delineated critical density thresholds for various infrastructures, signifying points at which pedestrian behavior undergoes significant changes. According to Holl, densities below 0.7 ped/$m^{2}$ indicate a free flow state, densities below 1.3 ped/$m^{2}$ represent a bound flow, and values above 1.3 ped/$m^{2}$ are indicative of congested flow. In our model, we refine these categories by slightly narrowing the bound flow range and subdividing the congested flow category into two distinct segments.

Figure 3 illustrates the procedural steps undertaken to evaluate our proposed methodology and within the sizes of the clusters, which are partly taken from the classification Figure 2.

New trajectory scenes are given to our framework, where the initial step involves calculating the scene’s density using Equation (Equation 12) in individuals per square meter (ped/$m^{2}$), *N* represents the total number of pedestrians observed within the scene, and *A* is the scene’s total area in square meters.

The density categorization is as follows: scenes with a density below 0.7 ped/$m^{2}$ are labeled as lowD; densities ranging from 0.7 to 1.2 ped/$m^{2}$ are classified as mediumD; densities between 1.2 and 1.6 ped/$m^{2}$ are designated as highD; and densities exceeding 1.6 ped/$m^{2}$ are identified as veryHD. Following classification, the scene is bifurcated into two segments: the initial segment spans nine timesteps and serves as input for one of the four specialized Sub-LSTMs, while the subsequent segment, encompassing twelve timesteps, is utilized to appraise the LSTMs’ performance through the computation of error metrics ADE, FDE, and COL. Figure 4a–d provide illustrative examples of each density level encountered in our dataset.

In Figure 4a, the scene exhibits very low density, with pedestrian movement primarily from two directions, leading to numerous interactions and avoidance behaviors. This is characteristic of our lowD data. Conversely, Figure 4b showcases a moderately higher density, yet still affords space for interactions, avoidance, and bidirectional pedestrian flow. In Figure 4c, representing highD data, the dynamics of pedestrian movement markedly differ from those observed in lowD and mediumD scenes, with movement predominantly unidirectional from the top, indicating a tendency to follow the pedestrian ahead. This pattern is even more pronounced in Figure 4d, where the flow from the top is so dense that passage from the bottom becomes challenging, leading pedestrians to follow the leader with limited freedom of movement and space. These observed behavioral differences underpin our classification rationale.

### 4.4. Collision Weight

Predictions of pedestrian trajectories presents the challenge to predict trajectory paths that do not collide with neighbours. Accurately measuring these collisions is challenging due to the shapes of pedestrians, which can vary from person to person. Traditionally, collisions are defined by the overlap of the radii of two pedestrians, as delineated in Equations (Equation 3)–(Equation 5). However, this method proves suboptimal for inclusion as a penalty function within the loss function of DL algorithms. Analysis of pedestrian trajectory data frequently reveals instances where collisions are not genuine but rather instances of grouping behavior, with individuals walking closely together, sometimes shoulder-to-shoulder. It is not these interactions we aim to deter, but rather scenarios in which individuals move directly towards one another without any attempt to avoid collision—behaviors that are unrealistic and undesirable.

To address this, we adopt the TTC concept, a widely recognized principle in the study of pedestrian dynamics [7]. Implementing this variable in the loss function of an DL algorithm would reduce predicted situations, where pedestrians walk straight towards each other without avoidance mechanism. Integrating TTC into a DL algorithm’s loss function significantly mitigates predictions where pedestrians are on a direct collision course without any avoidance mechanisms. The TTC term calculates the time until two pedestrians would collide if they continue moving at their current velocities, a concept validated by Karamouzas et al. [7]. The relative position and velocity between the pedestrian *i* and *j* can be denoted by $x_{ij}=x_{i}-x_{j}\in R^{2}$ and $v_{ij}=v_{i}-v_{j}\in R^{2}$, respectively. A collision between pedestrian *i* and pedestrian *j* occurs if a ray, originating from $\left(x_{i},y_{i}\right)$ and extending in the direction of $v_{ij}$, intersects the circle centred at $\left(x_{j},y_{j}\right)$ with a radius of $R_{i}+R_{j}$ at some time $\tau_{ij}$ in the future. This condition can be mathematically represented as $\left|\right|x_{ij}+v_{ij}.t{\left|\right|}^{2}<{\left(R_{i}+R_{j}\right)}^{2}$, where ||.|| denotes Euclidean norm. Solving this quadratic inequality for *t* yields $\tau_{ij}$ as the smallest positive root:(14)$$\tau_{ij}=\frac{-x_{ij}\cdot v_{ij}-\sqrt{{\left(x_{ij}\cdot v_{ij}\right)}^{2}-\left|\right|v_{ij}{\left|\right|}^{2}\left(|\right|x_{ij}{\left|\right|}^{2}-{\left(R_{i}+R_{j}\right)}^{2})}}{\left|\right|v_{ij}{\left|\right|}^{2}}$$

A collision is imminent when $\tau_{ij}=0$, whereas a large positive value for $\tau_{ij}$ indicates no collision risk. To implement $\tau_{ij}$ into the loss function, we have to use an sigmoid function *f* that has high values, if $\tau_{ij}$ is low and vice versa:(15)$$f\left(\tau \right)=\frac{1}{1+e^{s(\tau -\delta )}},$$
where s=10 and δ=0.4 are slope and threshold parameters, respectively. This function is then integrated into the loss function, traditionally focused solely on minimizing the ADE. The revised loss function combines ADE with TTC loss, optimized through minimization:(16)$$L_{i}=\sum_{t=1}^{T}∥x_{i}\left(t\right)-{\hat{x}}_{i}\left(t\right)∥+\lambda \sum_{t=1}^{T}f\left(\min_{j\neq i}\left\{\tau_{ij}\right\}\right),$$
where λ>0 modulates the influence of the TTC component in their loss function. The calculation of $\tau_{ij}$ considers all nearby pedestrians to the primary pedestrian, employing the minimum $\tau_{ij}$ to identify and mitigate the most critical potential collision scenario in the model.

### 4.5. Implementation Details

The algorithms are implemented in the commonly accepted configurations of related contributions [23]. All computations are performed using the PyTorch framework. The learning rate is set to 0.001 and an ADAM optimizer is utilizied. The batch size is set to 8 and training is carried out for 15 epochs, if not the early stop mechanism interrupt. This is the case, when the validations error starts to rise for three epochs. For validation and testing, a hold-out validation strategy is adopted by allocating 15% of the dataset for each validation and testing, while the remaining data serves as the training set. For capturing pedestrian interactions, we choose a circles with a radius of r=5 m surrounding the primary pedestrian.

## 5. Results

We will unveil the predictive outcomes of our dataset using two distinct yet synergistic methods. Initially, we will showcase the performance of our two-stage prediction framework, comparing it with contemporary state-of-the-art methodologies. Subsequently, we will demonstrate the seamless integration of our two-stage process with the incorporation of the TTC term into the loss function, illustrating its efficacy in mitigating collision instances.

### 5.1. Two-Stage Predictions

The results of our predictions will be presented in Table 1. As described in Section 4.3, we evaluated the predictions on the different density levels lowD, mediumD, highD, and veryHD. For every approach, we measure ADE, FDE, and COL metrics.

The initial insight gleaned from Table 1 reveals a clear trend: as density increases, the COL metric rises while the ADE/FDE diminish. This pattern emerges because higher densities naturally lead to reduced distances between individuals, consequently resulting in increased overlaps among agents. Additionally, it is observed that velocities decrease as density intensifies, leading to trajectories that are shorter in spatial extent. This reduction in travel distance directly contributes to the observed decrease in both ADE and FDE metrics at higher densities. Furthermore, it is clear that the DB algorithm outperform the traditional models CV and SF in terms of ADE/FDE. In terms of the COL metric, SF performs very well. In the last three rows of Table 1, we present the effectiveness of our two-stage approach and its combination with the TTC term. The results clearly show a significant improvement in the algorithm’s precision, attributed to the strategy of classification before prediction. Our enhanced two-stage SLSTM model consistently outperforms the traditional SLSTM across all evaluated datasets, demonstrating superior performance in terms of ADE, FDE, and COL metrics. Similarly, our adapted SGAN model shows marked improvements over the standard SGAN in three out of four datasets with respect to ADE. Integrating the TTC term further enhances the SLSTM results, notably in reducing collisions. A more detailed discussion on this enhancement is provided in the subsequent Section 5.2.

### 5.2. Collision Weight

In this study, we propose to integrate TTC in the loss function with the two-stage approach outlined in Section 4.3. As described in Equation (Equation 16), the collision part in the loss function can be adjusted by a parameter λ [36]. If λ is high, the impact of the TTC term is high compared to the impact of ADE and vice versa. In Figure 5, the impact of different values of λ on the prediction accuracy of the SLSTM algorithm is displayed: Figure 5a for lowD data, Figure 5b for mediumD data, Figure 5c for highD data, and Figure 5d for veryHD data. The first observation in each Figure is for λ=0, which means that it is equivalent to the value of our two-stage SLSTM in Table 1.

Across all figures, a consistent pattern emerges: increasing the collision weight λ generally results in fewer collisions. Optimal predictions occur at a specific λ value, where ADE and FDE are equivalent to or lower than those at λ=0. Each dataset exhibits a maximum effective λ value, beyond which ADE sharply increases. In the lowD dataset (Figure 5a), the ideal λ is 0.08, reducing ADE by 19% and collisions by 6%. Values slightly higher than 0.08 are still beneficial, yielding fewer collisions and enhanced avoidance behavior, but λ values exceeding 0.16 lead to a significant increase in ADE. In the mediumD dataset (Figure 5b), the optimal λ is 0.04, reducing ADE and collisions by 3% and 40%, respectively. Here, a λ value above 0.06 results in increased ADE, although a λ of 0.1 reduces the collision metric by 75%. In the highD dataset, improvements in ADE are marginal too, only notable at a λ of 0.02. However, the collision metric significantly decreases, by up to 48%, at a λ of 0.08. Conversely, in the very high-density (veryHD) dataset, increasing collision weight initially results in a rise in ADE, with no subsequent decrease. While the collision metric decreases by 37% at λ=0.08 values, there are no improvements for ADE. These empirical observations lead to the insight, that pedestrian behavior at different densities is very different and need different parameter configurations. At lower densities, our TTC term can improve overall accuracy (ADE and COL), while at higher densities, we can only reduce COL, by taking higher ADE into account.

## 6. Conclusions

Pedestrian behavior is inherently complex, exhibiting a wide variety of patterns across different contexts. This paper introduced a novel pedestrian trajectory dataset, characterized by its diversity in situational contexts, including varying densities and motivations. Our analysis of the data reveals variations in pedestrian behaviors correlating with the density of the scene. To address these variations, we propose a novel two-stage classification and prediction process. This approach first classifies scenes based on density and then applies the suitable model for predicting behavior within that specific density context. Implementing this framework enhanced the prediction accuracy of two famous DL algorithms, Social LSTM and Social GAN.

Furthermore, we integrated a TTC-based term into the loss function of the SLSTM to improve avoidance behaviors, consequently reducing potential collisions. Our empirical studies indicate that the effectiveness of the TTC-based term varies with density; it significantly benefits scenarios of low density by correlating higher TTC values with reduced collision incidents. However, the outcomes in high-density situations were more ambiguous, suggesting a nuanced impact of density on the efficacy of this approach. This observation could be attributed to the nuanced dynamics of pedestrian behavior across different densities. Specifically, in environments with lower densities, pedestrians tend to navigate more through avoidance and interactions, making TTC particularly relevant. Conversely, in higher density settings, pedestrian movement is more characterized by forced leader–follower dynamics, diminishing the prominence of TTC in explaining behavior. This study underscores the complexity of pedestrian behavior, which varies significantly under different environmental conditions. It highlights the necessity of adopting a flexible modeling approach to accurately predict pedestrian trajectories in diverse settings.

This research opens several avenues for future investigation in the field of pedestrian trajectory prediction, especially concerning heterogeneous datasets characterized by variable densities. Current methodologies typically rely on a one-size-fits-all model for behavior prediction across all conditions. We advocate for the development and application of multiple specialized models, each tailored to different scene characteristics, with scene density being a pivotal factor. While focusing on density has proven to be a successful strategy, exploring additional factors and individual pedestrian behavior characteristics could yield further improvements. For instance, our methodology utilized an estimate of overall scene density. However, pedestrians do not take global densities for their decision making into account, but rather local densities. Wirth et al. [41] demonstrated that pedestrian decisions are primarily influenced by their visual neighborhood. Future studies should investigate the impact of assessing local density variations within a scene, which could be particularly beneficial in environments exhibiting a wide range of density levels. This direction could unlock new dimensions of accuracy and reliability in trajectory prediction models.

Moreover, incorporating the TTC concept into the loss function has shown promise in enhancing prediction accuracy at lower density levels. Future research should explore alternative loss functions, particularly for high-density scenarios, where the traditional ADE-based approaches may not suffice. Investigating other metrics that could more effectively capture the complexities of high-density pedestrian behavior is crucial for advancing the field. 

## Figures and Tables

**Figure 1 sensors-24-02356-f001:**
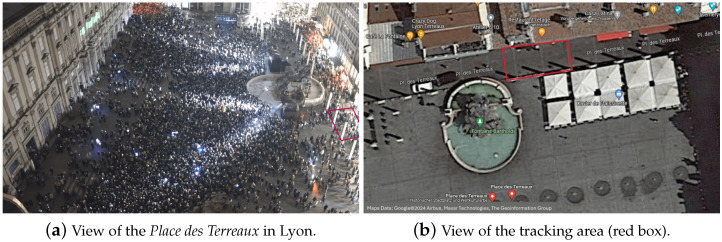
Area for the trajectory tracking at Lyon’s Festival of Lights 2022.

**Figure 2 sensors-24-02356-f002:**
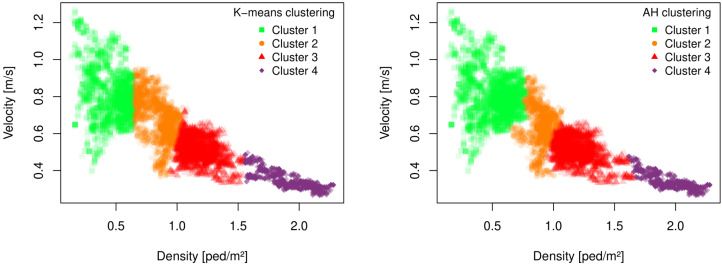
Results of the K-Means and the agglomerative hierarchical clustering. Trajectory scenes are clustered as shown by the different colors of the points.

**Figure 3 sensors-24-02356-f003:**
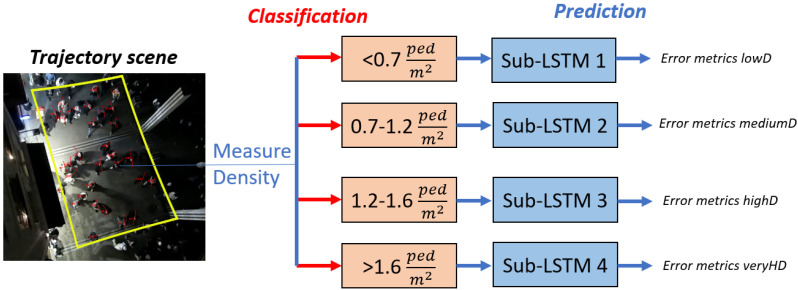
Schemata of our two-stage prediction approach.

**Figure 4 sensors-24-02356-f004:**
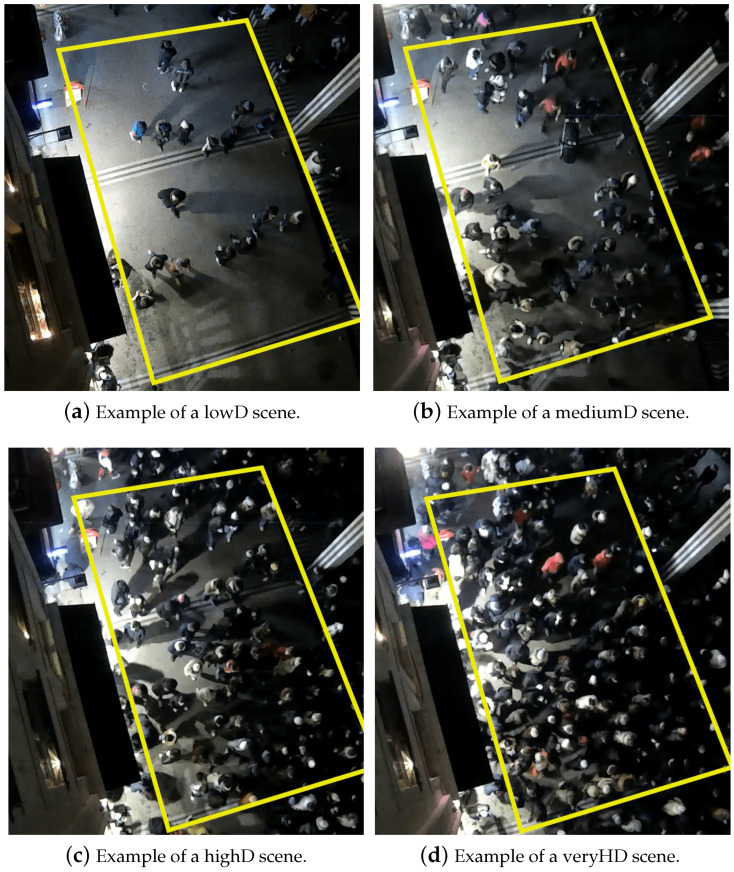
Examples for each of the four density levels.

**Figure 5 sensors-24-02356-f005:**
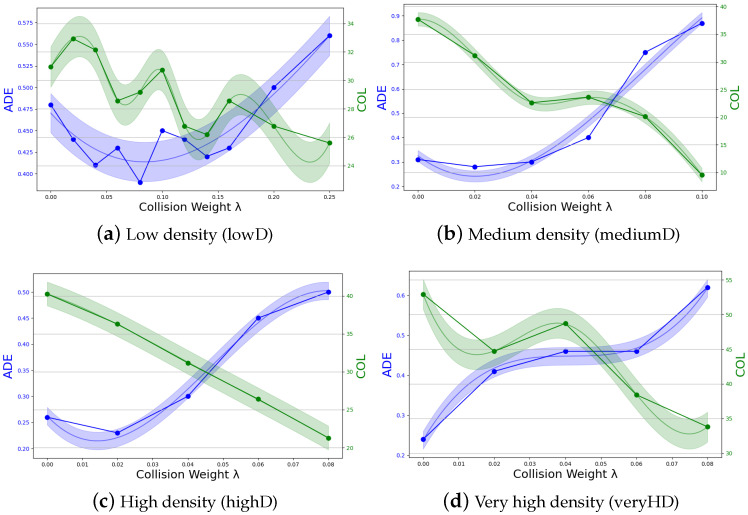
ADE and COL metrics for the two-stage SLSTM algorithm according to the collision weight λ for the four density levels.

**Table 1 sensors-24-02356-t001:** Quantitative comparison of ADE, FDE and COL prediction metrics for classical algorithms borrowed from the literature and our 2-stage approaches on the Festival of Lights dataset.

Model	LowD		MediumD		HighD		VeryHD	
	ADE/FDE	COL	ADE/FDE	COL	ADE/FDE	COL	ADE/FDE	COL
CV	0.71/0.97	54.76	0.85/0.98	45.73	0.53/0.8	62.35	0.44/0.67	81.74
Social Force [8]	0.78/1.33	**24.4**	0.55/0.89	**31.16**	0.5/0.82	36.43	0.36/0.63	54.78
Vanilla LSTM	0.5/0.99	31.55	0.33/0.63	37.69	0.29/0.52	36.43	**0.24/0.41**	63.8
Social LSTM [5]	0.53/1.02	57.74	0.37/0.73	59.3	0.41/0.78	64.26	0.35/0.66	75.37
Social GAN [14]	0.53/0.99	31.36	0.39/0.72	32.16	0.36/0.61	**32.33**	0.25/0.41	55.94
**Our 2stg. SLSTM**	0.48/0.93	30.95	**0.3/0.63**	36.18	**0.26/0.4**	42.02	**0.24/0.41**	**52.23**
**Our 2stg. SGAN**	**0.44/0.83**	32.74	**0.27/0.52**	40.2	0.28/0.5	**35.33**	0.26/0.43	58.6
**Our 2stg. TTC-SLSTM**	**0.39/0.73**	**29.17**	**0.3/0.62**	**22.61**	**0.23/0.36**	36.29	**0.24/0.41**	**52.23**

## Data Availability

The data are available on the following website: https://madras-data-app.streamlit.app/ (accessed on 1 April 2024). For more information see: https://www.madras-crowds.eu/.

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
