# Peer review of "Toward Better Pedestrian Trajectory Predictions: The Role of Density and Time-to-Collision in Hybrid Deep-Learning Algorithms"

_sensors, 2024, doi:10.3390/s24072356_

Round 1

Reviewer 1 Report

Comments and Suggestions for Authors

In this work the authors present a novel data-driven approach for predicting pedestrian trajectories. They compare their approach with a few established methods from the literature on a real-life dataset, comparing the accuracy obtained for various level of density. I really enjoyed reading this work, the quality of the presentation is excellent and very detailed, and I particularly appreciated the comparison with different methods. I only have a few minor comments:

1) It would be good to add a few more details on the dataset, maybe in an appendix section. What is the typical lenght of a trajectory? Would be nice to see a PDF of the trajectory lenght, and some other figures reporting what kind of flows are observed (mostly bidirectional?). Moreover it would be very helpful to show examples in which the prediction of SLSTM/SGAN are off from the ground truth.

2) In their review of trajectory datasets for pedestrian, the authors mentiod that most of those available in the literature cover only the "free-stream" regime (< 0.5 ped/m^2). The authors might be interest in taking into consideration the following dataset https://doi.org/10.5281/zenodo.7007358 which seems to take into account scenarios closely related to the one object of the present study. Would the networks trained be able to make predictions on other datasets as well? Tests and comments on the generalization capabilities of the algorithm here presented would help conveying a much more stronger message in my opinion.

Author Response

1) It would be good to add a few more details on the dataset, maybe in an appendix section. What is the typical lenght of a trajectory? Would be nice to see a PDF of the trajectory lenght, and some other figures reporting what kind of flows are observed (mostly bidirectional?). Moreover it would be very helpful to show examples in which the prediction of SLSTM/SGAN are off from the ground truth.

Response: Thank you very much for this valuable comment. To enhance the reader's understanding of our data, we have included additional figures in the appendix (Figures A1-A3), which cover, among other aspects, the lengths of trajectories, mean speeds, and densities. Furthermore, Figure A4 provides examples of the algorithm's predictions, including the ground truth for better illustration.
We have also improved the description of our dataset in Section 3. For more information about our data we reefer to this website: \url{https://madras-data-app.streamlit.app/}.

2) In their review of trajectory datasets for pedestrian, the authors mentiod that most of those available in the literature cover only the "free-stream" regime $(< 0.5 ped/m^2)$. The authors might be interest in taking into consideration the following dataset \href{https://doi.org/10.5281/zenodo.7007358}{https://doi.org/10.5281/zenodo.7007358} which seems to take into account scenarios closely related to the one object of the present study. Would the networks trained be able to make predictions on other datasets as well? Tests and comments on the generalization capabilities of the algorithm here presented would help conveying a much more stronger message in my opinion..

Response: Thank you for bringing this exciting data set to our attention. We have incorporated it into Section 3 of our text. While the trajectory lengths are too short for integration into our existing framework, we recognize its potential value. We intend to explore opportunities to utilize this dataset in future research endeavors.

Reviewer 2 Report

Comments and Suggestions for Authors

The prediction of pedestrian trajectories has become increasingly accessible and intriguing with the advancement of video processing and computer technology. Developing robust models in this field could potentially benefit various application areas. This study proposes a novel two-stage method utilizing different models to predict pedestrian trajectories. The writing style of the paper is clear and fluid; additionally, the provided tables and figures are meticulous and necessary. Since the references cited by the authors are generally relevant and up-to-date, it can be concluded that the paper is of high quality in terms of presentation.

The innovative aspect of the study lies in using different models for various pedestrian densities and convincingly demonstrating this approach. Furthermore, the novelty of the paper is enhanced by using image data containing high pedestrian density, unlike other datasets.

Upon examination of the methods, it is observed that various new machine learning techniques are employed. It would be appropriate to add a brief explanation of why these models were specifically chosen. It is also evident that sufficient performance metrics are used to evaluate the models' performance. While the K-means clustering algorithm used to cluster densities is widely known and validated, it is recognized that different clustering algorithms can yield different results. Hence, considering the results of different clustering algorithms could enhance the quality of the study.

In conclusion, the study shows promise for contributing to scientific advancement. I believe that the quality of the paper could be further improved by adding the points mentioned above. Congratulations to the researchers for conducting such a high-quality study.

Author Response

The prediction of pedestrian trajectories has become increasingly accessible and intriguing with the advancement of video processing and computer technology. Developing robust models in this field could potentially benefit various application areas. This study proposes a novel two-stage method utilizing different models to predict pedestrian trajectories. The writing style of the paper is clear and fluid; additionally, the provided tables and figures are meticulous and necessary. Since the references cited by the authors are generally relevant and up-to-date, it can be concluded that the paper is of high quality in terms of presentation.

The innovative aspect of the study lies in using different models for various pedestrian densities and convincingly demonstrating this approach. Furthermore, the novelty of the paper is enhanced by using image data containing high pedestrian density, unlike other datasets.

Upon examination of the methods, it is observed that various new machine learning techniques are employed. It would be appropriate to add a brief explanation of why these models were specifically chosen. It is also evident that sufficient performance metrics are used to evaluate the models' performance. While the K-means clustering algorithm used to cluster densities is widely known and validated, it is recognized that different clustering algorithms can yield different results. Hence, considering the results of different clustering algorithms could enhance the quality of the study.

In conclusion, the study shows promise for contributing to scientific advancement. I believe that the quality of the paper could be further improved by adding the points mentioned above. Congratulations to the researchers for conducting such a high-quality study.

Response: Thank you for your insightful comment and the congratulations. We selected SLSTM and SGAN for our study due to their prominence and widespread recognition within the field. Additionally, we opted for VLSTM because of its relative simplicity, providing valuable insights into the necessary complexity for effective performance. Our aim was not to limit our investigation to deep learning algorithms exclusively; hence, we incorporated two physics-based models into our benchmark to ensure a broader methodological perspective. To clarify our rationale and methodological diversity, we have included the following statement in Section 4.2 of our manuscript: 'These approaches, characterized by their diverse features, are commonly selected for comparison and serve as benchmarks that must be surpassed.'
Furthermore, changes in section 4.3 have been made. We added the results of the AHM cluster algorithm to Figure 2. Now it can be seen, that both cluster algorithms show similar results and the AHM classifies according to density too. 

Reviewer 3 Report

Comments and Suggestions for Authors

The reviewer comment is attached in a pdf file

Author Response

Thank you for examining our work so closely and providing detailed feedback. Attached, you can find our responses to your comments.

Round 2

Reviewer 3 Report

Comments and Suggestions for Authors

The authors have addressed all of my concern. The current version can be accepted.